# Molecular Characterization, Bioinformatic Analysis, and Expression Profile of Lin-28 Gene and Its Protein from Arabian Camel (*Camelus dromedarius*)

**DOI:** 10.3390/ijms20092291

**Published:** 2019-05-09

**Authors:** Sultan N. Alharbi, Ibtehal S. Alduhaymi, Lama Alqahtani, Musaad A. Altammaami, Fahad M. Alhoshani, Deema K. Alrabiah, Saleh O. Alyemni, Khulud A. Alsulami, Waleed M. Alghamdi, Mohannad Fallatah

**Affiliations:** 1National Center for Stem Cell Technology, King Abdulaziz City for Science and Technology, Riyadh 11442, Saudi Arabia; ialduhaymi@kacst.edu.sa (I.S.A.); lalqahtani@kacst.edu.sa (L.A.); mfallatah@kacst.edu.sa (M.F.); 2National Center for Biotechnology, King Abdulaziz City for Science and Technology, Riyadh 11461, Saudi Arabia; mtammami@kacst.edu.sa (M.A.A.); fhoshani@kacst.edu.sa (F.M.A.); wghamdi@kacst.edu.sa (W.M.A.); 3National Center for Pharmaceutical Technology, King Abdulaziz City for Science and Technology, Riyadh 11461, Saudi Arabia; dalrabiah@kacst.edu.sa (D.K.A.); saleh.o.alyemni@hotmail.com (S.O.A.); kaalsulami@kacst.edu.sa (K.A.A.)

**Keywords:** Arabian Camel, Lin-28, PMF-MS, 3D structure, cold-shock domain (CSD), bioinformatics analysis, phylogenetic analysis

## Abstract

Lin-28 is an RNA-binding protein that is known for its role in promoting the pluripotency of stem cells. In the present study, Arabian camel Lin-28 (cLin-28) cDNA was identified and analyzed. Full length cLin-28 mRNA was obtained using the reverse transcription polymerase chain reaction (RT-PCR). It was shown to be 715 bp in length, and the open reading frame (ORF) encoded 205 amino acids. The molecular weight and theoretical isoelectric point (pI) of the cLin-28 protein were predicted to be 22.389 kDa and 8.50, respectively. Results from the bioinformatics analysis revealed that cLin-28 has two main domains: an N-terminal cold-shock domain (CSD) and a C-terminal pair of retroviral-type Cysteine3Histidine (CCHC) zinc fingers. Sequence similarity and phylogenetic analysis showed that the cLin-28 protein is grouped together *Camelus bactrianus* and *Bos taurus*. Quantitative real-time PCR (qPCR) analysis showed that cLin-28 mRNA is highly expressed in the lung, heart, liver, and esophageal tissues. Peptide mass fingerprint-mass spectrometry (PMF-MS) analysis of the purified cLin-28 protein confirmed the identity of this protein. Comparing the modeled 3D structure of cLin-28 protein with the available protein 3D structure of the human Lin-28 protein confirmed the presence of CSD and retroviral-type CCHC zinc fingers, and high similarities were noted between the two structures by using super secondary structure prediction.

## 1. Introduction

The domesticated one-humped camel (*Camelus dromedarius*) belongs to the camelidae family. It plays a major role in the culture and way of life in the Middle East, not only because of its economic benefits in terms of milk and meat production, but also as an ideal model animal for medical study [1]. Although there has been an increasing number of studies on camels in recent years [2,3], structural and biomolecular research of Arabian camel proteins has been limited. The Arabian camel genome is composed of 74 chromosomes. The DNA sequence of camels reveals wide differences with other species, including human and mice; nevertheless, functional similarities remain. Therefore, studies of Arabian camel proteins offer a potentially viable target for human clinical applications.

Lin-28 is a conserved RNA-binding protein that is highly expressed in embryonic stem cells [4,5]. It plays several main cellular functions, including differentiation, development, and glucose metabolism, but its two main functions are the inhibition of let-7 microRNA and the regulation of target mRNA translation [6]. The Lin-28 protein consists of two main domains, an N-terminal cold-shock domain (CSD) and a C-terminal pair of retroviral-type CCHC zinc fingers [4], which act as both negative regulators of let-7 miRNA biogenesis and are post-transcriptional regulators of mRNA translation [7]. These two domains are highly conserved among mammals. Lin-28 can be localized in both the nucleus and cytosol of cells and interacts with the precursor of let-7 microRNA in order to prevent its maturation [8,9]. CSD, with its β-barrel structure, can bind single-stranded nucleic acids, therefore causing translational repression and activation of mRNA in undifferentiated and differentiated cells [10]. Lin-28 and the other heterochronic genes are unique among the developmental regulators of mammalian species, as they globally coordinate the relative timing of various developmental circumstances. It is unknown whether developmental timing mechanisms and several parts of developmental patterning are commonly conserved, or whether they show great differences among species [11].

In spite of the fact that the Lin-28 protein has been highly characterized in humans, *C. elegans* and drosophila [12,13], there are no reports of the Lin-28 protein from the Arabian camel (cLin-28). The main aim of the present study was to obtain the full-coding sequence of cLin-28 mRNA and to identify its amino acid sequence. Furthermore, The tissue-specific expression level of the cLin-28 protein was examined across 11 different Arabian camel tissues. We believe that this genetic and structural information will be a helpful source for the annotation of the Arabian camel genome. We also assume that the study of biochemical and biophysical properties of cLin-28 gene is likely to provide molecular insights into Arabian camel genome.

## 2. Results

### 2.1. Tissue-Specific Expression Profile of cLin-28 mRNA

The expression of cLin-28 mRNA was examined in 11 different tissues of Arabian camel (Figure 1). Specific primers (Table 1) were designed to amplify about 636 bp for cLin-28 and 190 bp for GAPDH genes (as an endogenous control). In addition, the level of expression of cLin-28 mRNA in the 11 different tissues was examined using qPCR. The qPCR specific primers were also designed to amplify 101 and 80 bp for cLin-28 and β-actin mRNAs, respectively. The maximum expression of cLin-28 mRNA was noted in the Arabian camel lung, heart, liver, and esophagus, followed by nearly equal expression in the kidney, and testis, whereas the lowest expression was noted in the spleen, small intestine, brain, and muscle tissues (Figure 2).

### 2.2. Characterization of the Full Coding cLin-28 Gene

The sequence suggested that the fragment has a length of 715 bp. This sequence characterized the first cLin-28 mRNA from the Arabian camel. Based on sequence homology, the full coding regions were compared with the corresponding regions from other counterparts. The sequence represents the first full coding region of the lin-28 gene from the Arabian camel and was submitted to the NCBI GenBank (accession number MK562530). The predicted amino acid sequence of cLin-28 mRNA was found to consist of an open reading frame (ORF) of 205 amino acid residues (Figure 3). The amino acid sequence was submitted to the gene bank (accession number XP_010992036). The BLAST analysis for the coding region of cLin-28 revealed that it shared high similarity (98.1–79%) with Lin-28 from other species (Figure 4).

### 2.3. Identification of cLin-28 Protein by Using Mass Spectrometry

The molecular analysis of the 205 amino acid sequence of the cLin-28 protein using the Geneious software [14] predicted that this protein has a molecular weight (Mwt) of 22.389 kDa and an isoelectric point (pI) of 8.50 (Figure 5a). Accordingly, for peptide mass fingerprint mass spectrometry (PMF-MS), the targeted protein spot obtained from the separation of proteins with two-dimensional polyarylamide gel electrophoresis (2-DE) was manually excised (Figure 5b) and subjected to MS analysis for the identification of the cLin-28 protein. Of the total trypsin-digested peptide mass of cLin-28 protein, 11 peptides, which covered 48% (Figure 5c) of the entire protein sequence, were hit in the NCBIprot database (containing 4540232 sequences) by using the Mascot peptide fingerprint search engine with cLin-28 protein (accession no. XP_010992036) with a score of 120 and *p* < 0.05.

The mass spectrum revealed several protonated ions [M + H]+ in the peptide fragments (Figure 5d). The ions at 1353.6800, 1155.5100, 1537.5300, 2240.1300, 721.4700, 889.4300, 1157.6300, 1378.8600, 705.380, 807.5000, and 1001.2400 were the 11 trypsin digested peptides corresponding to amino acids 2–15, 16–26, 27–41, 27–46, 42–46, 75–81, 85–94, 105–118, 122–127, 150–156, and 174–183, respectively. As shown in Table 2, the peptide mass profiles were retrieved from NCBIprot database search engine, and the amino acid sequence of each digested peptide was recognized from the sequence of cLin-28 protein from the extracted spot of this protein on the 2-DE gel. The PMF-MS results also aligned with those from other species; the second best matching protein had a score of 120 for alpaca (accession no. XP_006196877.1) Lin-28 protein. The third and fourth best matching proteins also scored with 120 for pig (accession no. ADK26463.1) and cattle (accession no. NP_001179986.1) Lin-28 proteins.

### 2.4. Amino Acid Composition and Homology of cLin-28 Protein

The biological activity of the cLin-28 protein depends on the nature of its amino acid composition. The cLin-28 protein contains 70 charged (40.88%), 59 hydrophobic (27.24%), 23 acidic (12.95%), 28 basic (17.41%), and 46 polar amino acids (22.15%). The complete amino acid analysis and chemical composition of the predicted protein are illustrated in Table 3. The instability index [15] of the cLin-28 protein was also measured to be 58.89, which classified the protein as unstable (>40). The aliphatic index and the grand average of hydropathicity (GRAVY) of cLin-28 were 54.83 and −0.579, respectively. Different methods were utilized to evaluate the hydrophilicity and hydrophobicity of the cLin-28 protein along its amino acid sequence (Figure 6). The Kyte and Doolittle method [16] provides a graphic visualization to track the hydrophobic and hydrophilic regions of cLin-28 protein relative to a universal midpoint line (Figure 6a). This scale identified the surface-exposed regions of the protein at its C-terminal. The four major regions above the midpoint line indicate internal regions, whereas, the seven major regions of the profile that lie below the midpoint line are external regions of the cLin-28 protein. In order to predict the major antigenic determinants of the cLin-28 protein, Hopp and Woods’ method [17] was used. The plot-generated prediction profile for the cLin-28 protein is shown in Figure 6b. The largest peaks correspond to amino acids 120–140, and amino acids 180–190 represent the most hydrophilic segments of the protein and are associated with the antigenic site. The deepest valley (amino acids 40–60) is associated with longest helical secondary structure element (Figure 7).

The flexibility of cLin-28 protein is important for its actions as a transcriptional factor. The Karplus and Schulz approch [18] was utilized to predict the structural flexibility of cLin-28 protein (Figure 6c). The flexibility analysis revealed that the N-terminal of the cLin-28 protein is more flexible than its C-terminal regions. Consequently, those flexible regions can be considered epitopic regions for this protein as they are constructed from surface amino acids rather than internal amino acids. Furthermore, B cell epitopes were predicted by using the Kolaskar and Tongaonkar antigenicity method [19] (Figure 6d). The average antigenic tendency value for this protein was 1.023 with a minimum value of 0.888 and a maximum value of 1.182. The results also showed that this protein has five possible antigenic peptides, with different lengths ranging from six to 33 amino acids. Amongst them, the two most preferred B cell epitope characteristics were observed for amino acids 58 to 74 and 143 to 175 (Table 4). In order to recognize the amino acids that are on the surface of the cLin-28 protein, Emini surface accessibility prediction [20] was utilized (Figure 6e). Amino acids located on a specific area in the cLin-28 protein can bind to the B cell receptor, and the area must be on the surface and immunogenetic. The maximum surface probability value for the cLin28 protein was found to be 5.957 from amino acid positions 115 to 135. In addition, Chou and Fasman β-turn prediction [21] predicts the epitope-based on the turn structure degree. It suggested that the epitope must always be found in the β-shaped turn structure. The results suggested that this protein is rich in β-turns in the region between amino acids 105 to 145, which is the region where β-strands are directed in anti-parallel to form β-sheets (Figure 6f).

### 2.5. Multiple Sequence Alignment

The amino acid sequence of the cLin-28 molecule was aligned with those of nine other mammalian species by ClustalW [22]. The two main domains—an N-terminal cold-shock domain (CSD) and a C-terminal pair of retroviral-type CCHC zinc fingers—were highly conserved for all aligned proteins. This protein binds to a pri- and pre-let-7 microRNA and represses their processing by Drosha and Dicer. Many biochemical and structural observations have revealed that the specificity of this interaction is mainly carried by the zinc finger domain with a conserved GGAGA motif. The cLin-28 protein shares an overall sequence identity of 80% and contains low-complexity regions at the N-terminus (Figure 8). The BlastP analysis [23] revealed that the cLin-28 protein shares high similarity with Lin-28 proteins from other mammalian species. The highest similarity was found with alpaca (98.6%), cattle (98.1%), pig (98.1%), Chinese hamster (96.2%), American beaver (96.2%), house mouse (95.7%), and horse (94.6%) proteins (Table 5 and Figure 8). In addition, multiple alignment analysis revealed that the RNA-binding domains of the Lin28 protein are highly conserved. This high resemblance suggests a close evolutionary relationship. The phylogenetic tree grouped this cLin-28 protein with wild bactrian camel and cattle (Figure 9) using the neighbor-joining method [24]. In order to predict the binding regions of the cLin-28 protein, the ANCHOR web-server [25] was used (Figure 10a). As shown in the figure, the N-terminal aspect of the cLin-28 protein is predicted to be the ordered region, whereas its C-terminal part is predicted to be the disordered region. The cLin-28 protein contains less disordered regions than the globular one (Figure 10b).

### 2.6. Secondary and 3D Structures of cLin-28 Protein

The secondary structure of the cLin-28 protein is a regular repeating folding pattern within this protein which is stabilized by hydrogen bonds between the amino and keto groups of the peptide bonds. The primary structure of the cLin-28 protein was utilized to predict its secondary structure, which revealed the first level of protein folding. The predicted structure demonstrated that the cLin-28 protein is composed of two α-helices and six β-strands (Figure 7), in which the six β-stranded structure forms a highly conserved CSD of about 80 amino acids (from 40 to 120 amino acids). A prediction of the secondary structure analysis of cLin-28 protein was carried out using PSIPRED server [26].

The activity and biochemical functions of the cLin-28 protein were determined by its three-dimensional (3D) shape. The 3D structure of the cLin-28 protein was predicted using homology structure modeling on the Phyre2 server [27]. Fold recognition with Phyre2 software retrieved the crystal structure of the human Lin-28 cold shock domain as the top hit with 99% confidence and 80% identity (PDB ID 4A4I). The predicted 3D structure of the cLin-28 protein showed overall folding and secondary structures that were very similar to those of the human Lin-28 protein (Figure 11a). The similarities between the cLin-28 protein and the human Lin-28 protein were studied by superimposing their structures in the PyMOL program (http://pymol.sourceforge.net) (Figure 11b). Overall, there were 0.417 RMSD deviation structure pairs.

The Ellipro server (http://tools.iedb.org/ellipro/) was utilized to predict the epitopic regions using the modeled 3D structure of the cLin-28 protein. The screening revealed that the cLin-28 protein has nine discontinuous with regions with score values of >0.5. The highest probability of a discontinuous epitope was calculated to be 80.3%. Table 6 details the amino acids involved in discontinuous epitopes, their sequence locations, the number of amino acids, and the scores. The higher the score is, the greater their potential of being a discontinuous epitope. Their positions of the epitopes on the 3D cLin-28 protein structure are shown in Figure 12.

## 3. Discussion

Although cDNA sequences are available for the Lin−28 gene of most species, previously, there were no reports of the full sequence, molecular characterization, and tissue distribution of the Arabian camel Lin−28 gene. This study presents the full-length cLin-28 cDNA sequence from the Arabian camel. The cLin-28 protein has a high degree of homology with other mammalian species and is phylogenetically clustered within the wild bactrian camel and cattle. The cLin-28 mRNA is 715 bp long. It contains an open reading frame of 615 bp that codes for 205 amino acids. The BLAST analysis and multiple sequence alignment of cLin-28 showed that the overall sequence homology of the cLin-28 protein sequence alignment with different species is high and shows similar structural characteristics.

We confirmed that the cLin-28 protein contains two highly conserved domains. The two main domains, an N-terminal cold-shock domain (CSD) and a C-terminal pair of retroviral-type CCHC zinc fingers, were highly conserved in all aligned proteins. The cLin-28 protein binds to a pri- and pre-let-7 microRNA and represses their processing by Drosha and Dicer. Many biochemical and structural observations revealed that the specificity of this interaction is mainly carried by the zinc finger domain with a conserved GGAGA motif. The cLin-28 protein shares an overall sequence identity of 80% and contains low-complexity regions at the N-terminus. Any single amino acid mutation in either the CSD or the retroviral-type CCHC zinc fingers eliminates both let-7 binding and processing inhibition. This indicates that both domains are required for Lin-28 function [28].

The species tree was constructed from the cLin-28 protein and nine of the highly similar mammalian Lin-28 proteins, with each species being represented by its own Lin-28 protein sequence. The internal nodes represent the ancestral sequences from which the present-day sequences have diverged after speciation events that produced two descendant divergent species. The species tree confirmed that the internal node of the Arabian camel protein has a further evolutionary distance from the root than the alpaca protein. The tree also demonstrated that the Wild Bactrian camel and cattle diverged from their common ancestor at a later evolutionary time than the Arabian camel.

We used the ANCHOR web-server to predict disordered regions of the cLin-28 protein. The 3D-structure of the cLin-28 protein partially lacks an ordered structure, especially at its termini regions. As a result, the cLin-28 protein can be classified as an intrinsically disordered protein. It has been suggested that all transcription factors tend to have fully or partially disordered regions in order to indicate that there is an intrinsic requirement for those transcription factors to be highly flexible and thus to be able to interact with other proteins and DNA [29]. The cLin-28 protein consists of the disordered N-terminal and C-terminal parts and the central largely ordered DNA binding domains.

## 4. Materials and Methods

### 4.1. Sample Collection

Eleven different Arabian camel tissues were obtained from an adult male camel slaughtered at the main slaughter-house in Southern Riyadh. Those tissue types used were spleen, lung, small intestine, heart, liver, kidney, stomach, esophagus, brain, muscle, and testis. All tissue samples were then immediately submerged in RNAlater^®^ solution (Qiagen, Hilden, Germany) to avoid RNA degradation and stored at 4 °C for 24 h and then at −20 °C until use. The other sample tissues were transported on ice to the laboratory to be utilized for proteomic analysis.

### 4.2. Isolation of RNA and Synthesis of cDNA

Total RNA was isolated from about 500 mg of each tissue under study using the Trizol Reagent method [30], as described in the manufacturer’s instructions (Invitrogen, Carlsbad, CA, USA). The concentration of obtained RNA was measured by the Nanodrop spectrophotometer (NanoDrop; TermoScientifc) at 260 nm, and the integrity and quality of isolated RNA was evaluated using denaturing agarose gel (1%) electrophoresis. Aliquots of 2 μg RNA samples were reverse-transcribed into first strand cDNA with a High-Capacity cDNA Reverse Transcription Kit using the Oligo (dT) primer in compliance with the standard protocol for PCR amplification.

### 4.3. Examining Clin-28 Gene Expression by Using PCR and qRT-PCR

The data from the Arabian camel genome project (http://camel.genomics.org.cn/page/camel/index.jsp) were used to design PCR and qPCR primers (Table 1). The PCR reaction mixture ( 25 μL) contained 12.5
μL of 2X GoTaq^®^ Green Master Mix (Promega, Madison, WI, USA), 1 μL of 5 pmol of each primer, 5 μL of template cDNA and was carried out by ProFlex PCR System (Applied Biosystems). The PCR amplification conditions were as follows: initial denaturation at 94 °C for 5 min, followed by 30 cycles at 59 °C for 5 s, 60 °C for 30 s, and 72 °C for 45 s. The final extension step was performed at 72 °C for 10 min. The PCR products were then electrophoresed on 1.2% agarose gel stained with SYBR^®^ Safe. The obtained PCR fragments were then sequenced according to the Sanger method [31] by using the 3730XL series platform sequencer. In addition, the level of relative expression of cLin-28 mRNA was measured by examining the 11 different Arabian camel tissues with the fluorescent quantitative real-time PCR (qRT-PCR) method. The GAPDH mRNA was used as an endogenous control. In this experiment, the Fast SYBR^®^ Green Master Mix kit was used, and gene-specific primer pairs were designed to amplify 101 bp of cLin-28. The qRT-PCR reaction mixture included 10 μL of Fast SYBR^®^ Green Master Mix (Cat. No., 4385612, Applied Biosystems), 1 μL of the forward primer, 1 μL of the reverse primer, 3 μL of nuclease-free water, and 5 μL of cDNA target to give a total volume of 20 μL. The thermal cycling parameters were as follows: initial denaturation at 95 °C for 3 min, and amplification of 40 cycles at 95 °C for 3 s and then at 60 °C for 40 s. The threshold cycle ΔΔ-Ct values for the cLin-28 gene and the GAPDH housekeeping gene were determined in triplicate. Relative quantification of RNA was calculated by the ΔΔ-Ct method.

### 4.4. Protein Digestion and Mass Spectrometry

The two-dimensional gel electrophoresis (2-DE) coupled with liquid mass spectrometry (LC-MS) was used to identification of cLin-28 protein. This procedure includes the separation of clin-28 protein based on its isoelctric point (pI) in the first dimension followed by separation on the basis of its molecular weight using sodium dodecyl sulfate polyacrylamide gel electrophoresis (SDS-PAGE). 25 μg of Arabian camel protein mixtures were separated by 2-D SDS-PAGE followed by staining of the gel with protein specific dye solution (Coomassie R-240) overnight. After destining, the cLin-28 protein spot (pI of 8.50 and Mwt of 22.389 kDa) was then digested as described before [32]. The tryptic-cleaved peptides mixture was transferred to a clean autosampler vial. Millipore^®^Ziptips C18 pipette (Tip size:P10, Merck KGaA, Darmstadt, Germany) was performed in order to prepare sample for Matrix-assisted laser desorption/ionization-time of flight (MALDI-TOF) mass spectrometry. 1 μL of aliquots were generally sampled directly from the digest supernatant for MS fingerprint analysis by using Axima Performance 1 MALDI TOF/TOF Mass Spectrometer (Shimadzu Corporation, UK). The output data were then searched using the MASCOT search engine (http://www.matrixscience. com).

### 4.5. Structure Modeling

The secondary structure of cLin-28 protein sequence was generated using Geneious software v10.0.3 [14]. The Phyre2 server (http://www.sbg.bio.ic.ac.uk/phyre2/html/page.cgi?id=index) was utilized to predict the three-dimensional (3D) structure of cLin-28 protein. The structural alignment between modeled clin-28 protein and human Lin-28 protein (PDB:4A4I) was performed by using Pymol software [33]. The quality of the structural alignment was evaluated using PDBe on (https://swissmodel.expasy.org/interactive).

## 5. Conclusions

In conclusion, 715-bp was sequenced and confirmed by using 3730XL series platform sequencer. The predicted amino acid sequence of cLin-28 mRNA was found to consist of an open reading frame (ORF) of 205 amino acid. The molecular analysis of the 205-amino acid sequence of cLin-28 protein using the Geneious software predicted that this protein has a molecular weight (Mwt) of 22.389 kDa and Isoelectric point (pI) 8.50. Peptide mass fingerprint-mass spectrometry (PMF-MS) analysis of the purified clin-28 protein confirmed the identity of this protein. Expression of cLin-28 protein was examined in several Arabian camel organs, with highest expression occurring in lung, heart, liver, and esophagus. A BLASTp search engine revealed that 205 amino acid translated protein of Arabian camel shared high similarity (98.1–79%) with Lin-28 from other species. The instability index of cLin-28 protein was also measured to be 58.89, which classified the protein as unstable. The flexibility analysis revealed that the N-terminal of cLin-28 protein is more flexible than its C-terminal regions. Sequence similarity and phylogenetic analysis showed that the cLin-28 protein is grouped together with those of Camelus bactrianus and Bos taurus. The predicted 3D structure of cLin-28 protein shows overall folding and secondary structures very similar to those of humans.

## Figures and Tables

**Figure 1 ijms-20-02291-f001:**
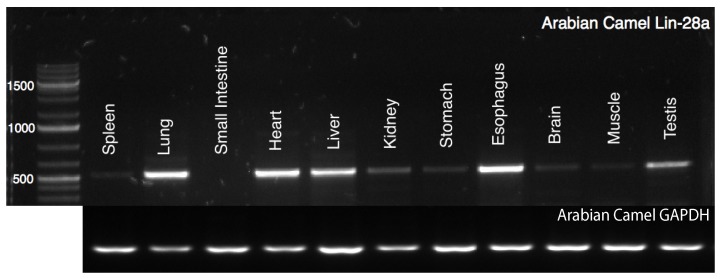
Agarose gel (1.2%) electrophoresis of PCR products for cLin-28 mRNA, 1500 bp DNA molecular weight marker was used.

**Figure 2 ijms-20-02291-f002:**
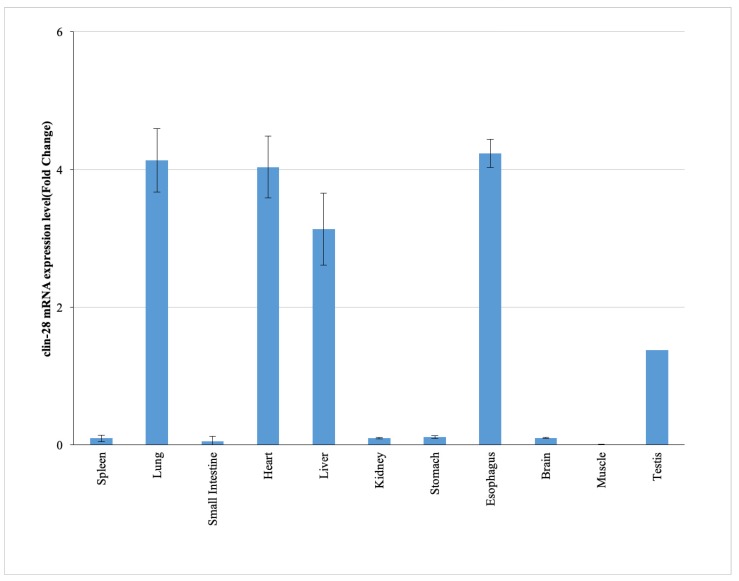
cLin-28 mRNA expression levels in different tissues. The results are expressed relative to an endogenous control, β-actin.

**Figure 3 ijms-20-02291-f003:**
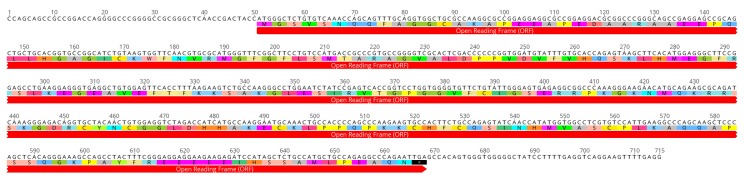
The nucleotide sequence and the deduced amino acids of Arabian camel Lin-28. The sequence was submitted to NCBI GenBank (accession number MK562530).

**Figure 4 ijms-20-02291-f004:**
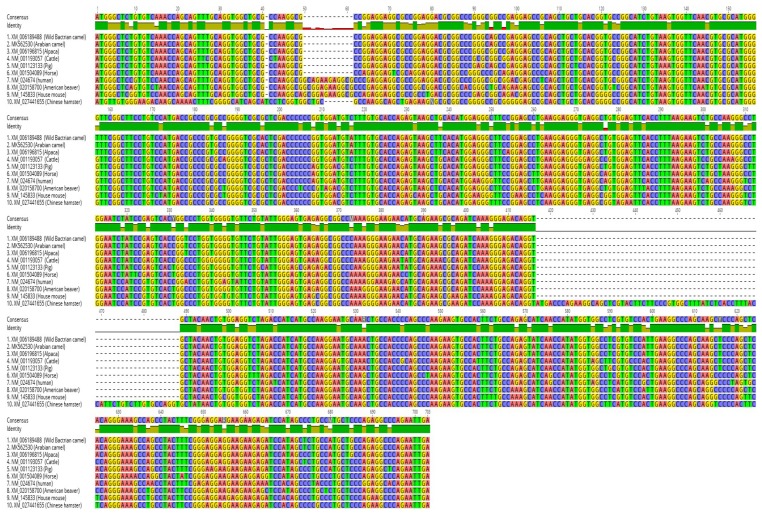
Multiple alignment of the nucleotide sequence of cLin-28 mRNA with nine other mammalian species. Identical nucleotides are marked in green.

**Figure 5 ijms-20-02291-f005:**
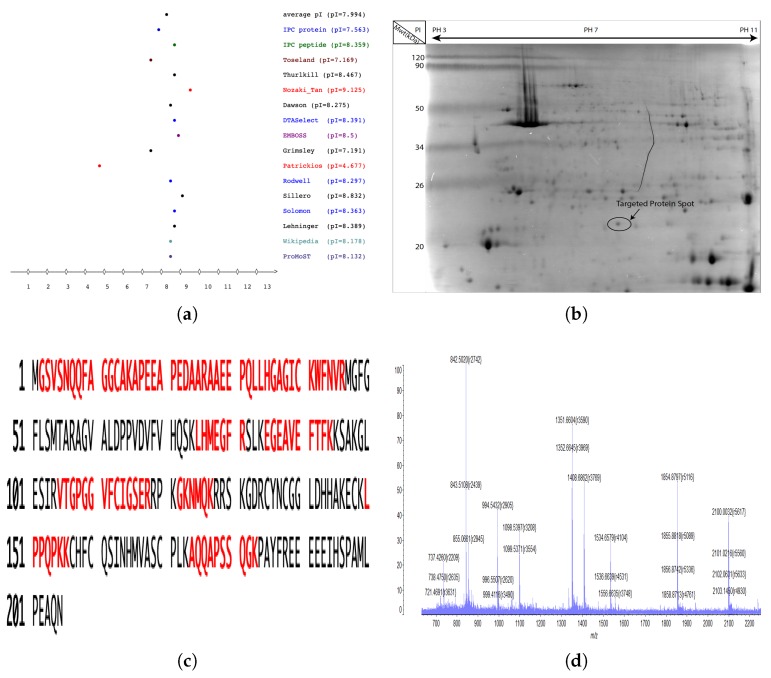
Liquid chromatography-mass spectrometry (LC-MS). (**a**) Isoelectric point (pI) of cLin-28 protein according to different scale calculations. (**b**) 2D-gel SDS page. (**c**) MLDI-TOF MS-derived peptides (red) matched to the sequence of cLin-28 protein. (**d**) Observed peak list of peptides from the cLin-28 protein.

**Figure 6 ijms-20-02291-f006:**
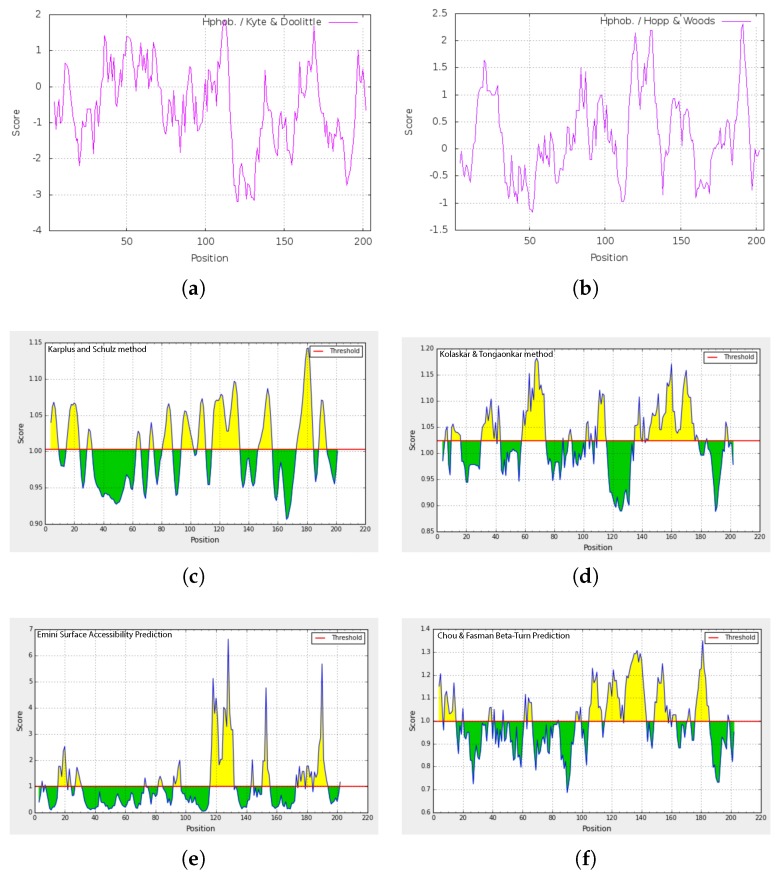
Hydrophatic profiles of the cLin-28 protein. Plots have amino acid sequence positions on the x-axis, and the degree of hydrophobicity and hydrophilicity on the y-axis. (**a**) Kyte–Doolittle method: the points above the midpoint line denote the hydrophopic regions and those below the midpoint line indicate hydrophilic regions of the cLin-28 protein. (**b**) Hydrophilicity plot for cLin-28 protein using the Hopp and Wood method. (**c**) Karplus and Schulz flexibility prediction of the cLin-28 protein. The x-axis and y-axis represent the position and score, respectively. The threshold is 1.0. The flexible regions of the protein above the threshold value are shown in yellow. (**d**) Kolashkar and Tongaonkar antigenicity prediction of the most antigenic regions of the cLin-28 protein. The threshold value is 1.0. The regions above the threshold are antigenic and are shown in yellow. (**e**) Emini surface accessibility prediction of the cLin-28 protein. The threshold value is 1.000. The regions above the threshold are antigenic and are shown in yellow. (**f**) Chou and Fasman β-turns prediction of clin-28 protein. The threshold is 1.00. The regions with β-turns in the protein are shown in yellow.

**Figure 7 ijms-20-02291-f007:**
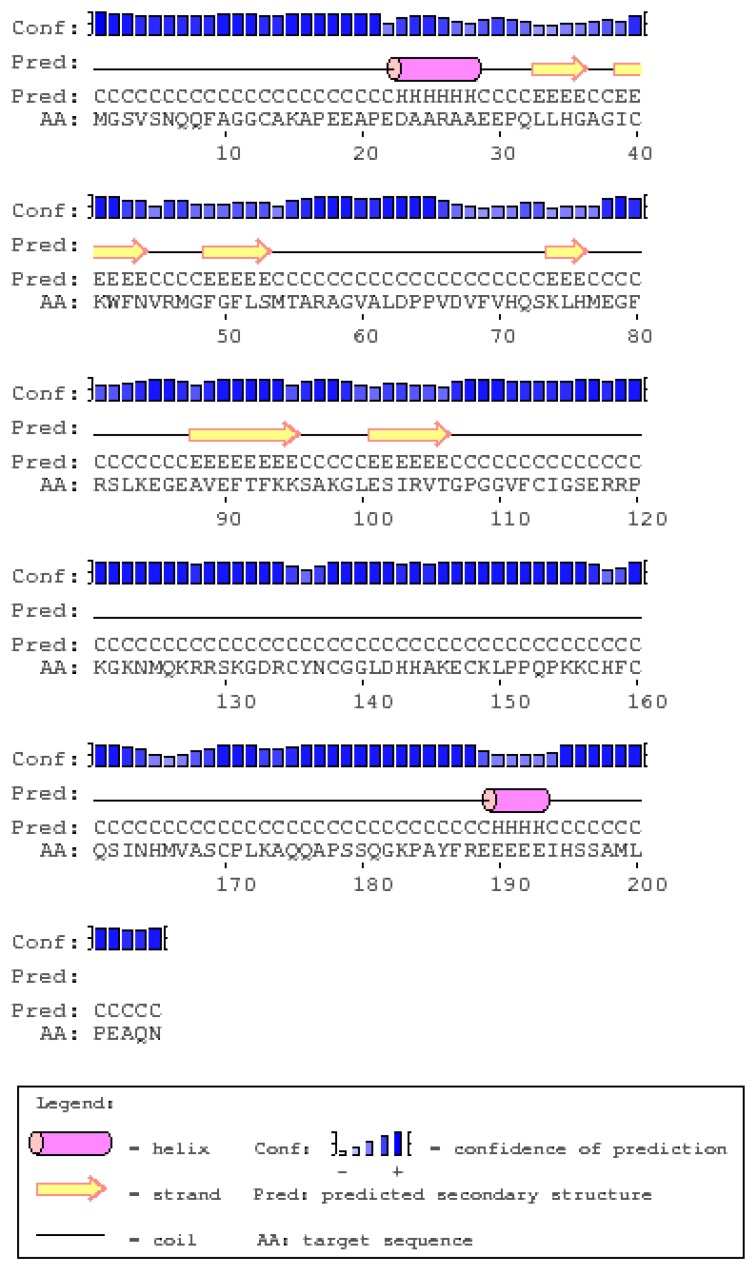
The secondary structure of the cLin-28 protein.

**Figure 8 ijms-20-02291-f008:**
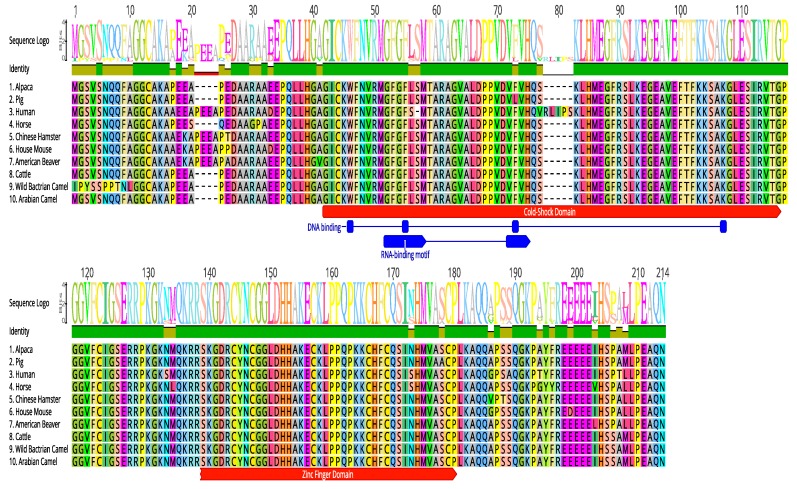
Multiple alignment of the amino acid sequence of cLin-28 protein with nine other mammalian species. Identical amino acids are marked in green, and typical cold shock and zinc finger domains are shown in red. RNA/DNA-binding domains are revealed in blue.

**Figure 9 ijms-20-02291-f009:**
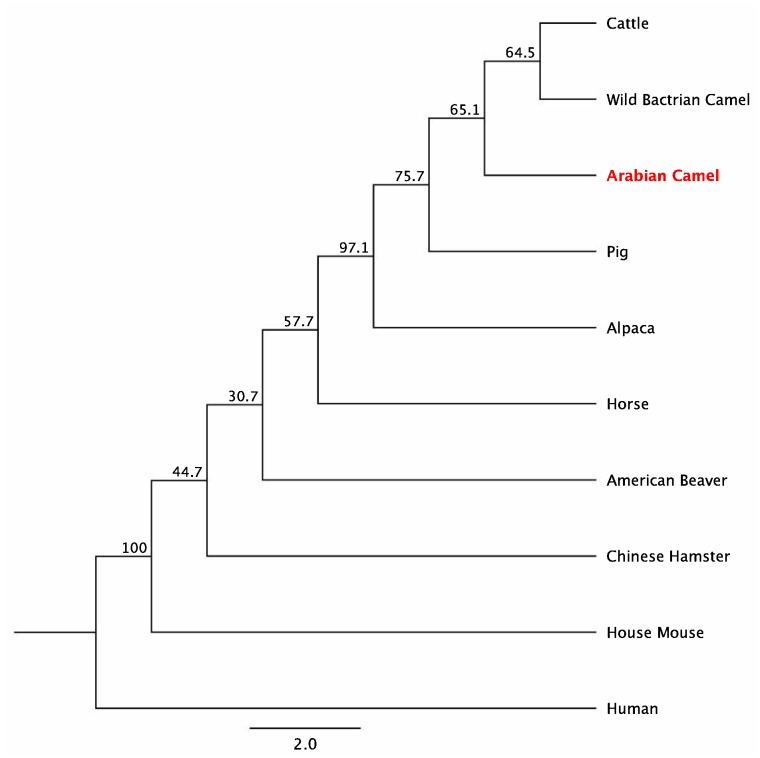
The phylogenetic tree shows the relationship of the cLin-28 protein with protein sequences from other species. The maximum likelihood tree is based on complete coding sequences. The values at nodes are bootstrapping 30% obtained from 1000-fold data re-sampling.

**Figure 10 ijms-20-02291-f010:**
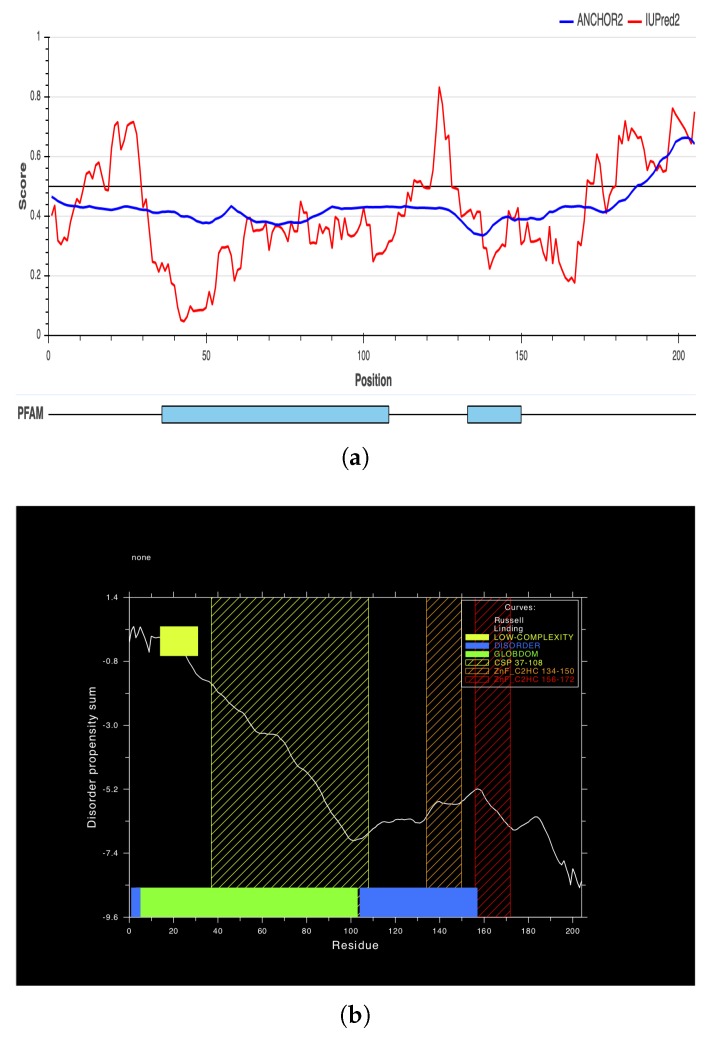
(**a**) Prediction of the binding regions of the cLin-28 protein. (**b**) The globular and disordered regions in the cLin-28 protein.

**Figure 11 ijms-20-02291-f011:**
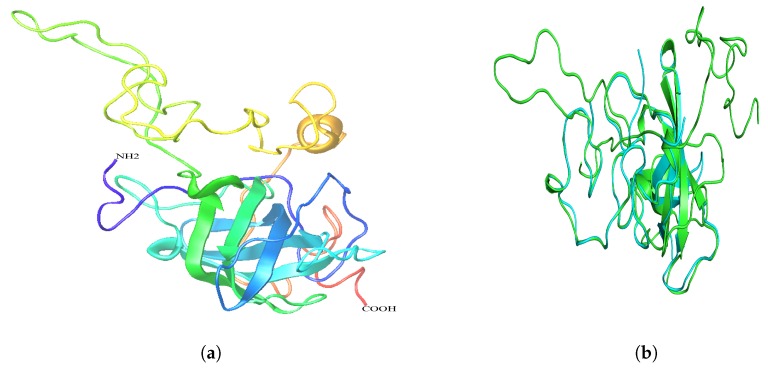
Modeled 3D structures. (**a**) The 3D structure of the cLin-28 protein. (**b**) Stereo ribbon representation of the predicted 3D structure model of the cLin-28 protein (cyan) and the superimposition with the human Lin-28 protein (green).

**Figure 12 ijms-20-02291-f012:**
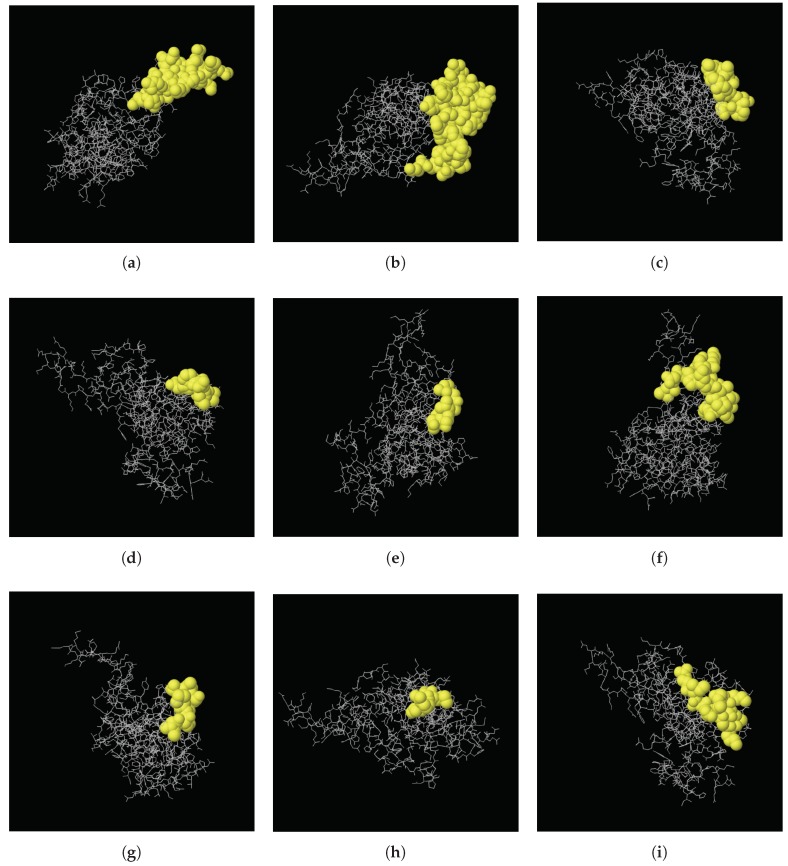
Three-dimensional representation of discontinuous epitopes (**a**–**i**) of cLin-28. The epitopes are shown in yellow, and the bulk of the cLin-28 protein is shown in grey sticks.

**Table 1 ijms-20-02291-t001:** List of primers used for the amplification and sequencing studies.

Usage	Primer Name	Primer Sequence 5${}^{\prime }$ → 3${}^{\prime }$	Length (bp)
ORF-PCR	cLin-28-F	ACTACCATGGGCTCTGTGTC	636
	cLin-28-R	ACCCACTGTGGCTCAATTCT	
	GAPDH-F	CTGGGAAGATGTGGCGTGAT	190
	GAPDH-R	AAGGCCATGCCAGTTAGCTT	
qRT-PCR	cLin-28-qF	AAAGCCAGCCTACTTTCGGG	101
	cLin-28-qR	CAAAAGGATAGCCCCCACCC	
	β-Actin-F	CCCATTGAGCATGGCATCGT	80
	β-Actin-R	GTAGATGGGCACAGTGTGAG	

**Table 2 ijms-20-02291-t002:** Observed and calculated ions of peptide masses of cLin-28 protein.

Amino Acid Positions	[M + H]+		
Start-End	B	Calculated (m/z)	Peptide Sequence
2–15	1353.6800	1353.4608	MGSVSNQQFAGGCAKA
16–26	1155.5100	1155.1723	KAPEEAPEDAARA
27–41	1537.5300	1536.7512	RAAEEPQLLHGAGICKW
27–46	2240.1300	2239.5544	RAAEEPQLLHGAGICKWFNVRM
42–46	721.4700	720.8185	KWFNVRM
75–81	889.4300	889.0332	KLHMEGFRS
85–94	1157.6300	1156.2416	KEGEAVEFTFKK
105–118	1378.8600	1378.5532	RVTGPGGVFCIGSERR
122–127	705.3800	704.8391	KGKNMQKR
150–156	807.5000	806.9923	KLPPQPKKC
174–183	1001.2400	1001.0520	KAQQAPSSQGKP

**Table 3 ijms-20-02291-t003:** Predicted chemical composition of the cLin-28 protein.

Amino Acid	Number Count	% By Weight	% By Frequency	
Ala (A)	21	6.67	10.24
Cys (C)	9	4.15	4.39
Asp (D)	5	2.57	2.44
Glu (E)	18	10.38	8.78
Phe (F)	11	7.23	5.37
Gly (G)	20	5.10	9.76
His (H)	8	4.90	3.90
Ile (I)	5	2.56	5.37
lys (K)	17	9.73	8.29
Leu (L)	11	5.56	5.37
Met (M)	7	4.10	3.41
Asn (N)	6	3.06	2.93
Pro (P)	14	6.07	6.83
Gln (Q)	11	6.30	5.37
Arg (R)	11	7.67	5.37
Ser (S)	15	5.83	7.32
Thr (T)	3	1.35	1.46
Val (V)	10	4.43	4.88
Trp (W)	1	0.88	0.49
Tyr (Y)	2	1.46	0.98

**Table 4 ijms-20-02291-t004:** Predicted antigenic peptides using the Kolaskar and Tongaonkar antigenicity method.

No.	Start	End	Peptide	Length
1	10	16	AGGCAKA	7
2	31	40	PQLLHGAGIC	10
3	58	74	AGVALDPPVDVFVHQSK	17
4	110	115	GVFCIG	6
5	143	175	HHAKECKLPPQPKKCHFCQSINHMVASCPLKAQ	33

**Table 5 ijms-20-02291-t005:** Homology of cLin-28 amino acids with those from other species.

Species	Common Name	Protein(Accession no.)	Protein Length (bp)	Identity (%)
*Homo sapiens*	Human	EAX07815.1	360	80
*Mus musculus*	House Mouse	AAH68304.1	209	95.7
*Sus scrofa*	Pig	ADK26463.1	205	98.1
*Camelus bactrianus*	Wild Bactrian Camel	EPY77519.1	232	88.6
*Camelus dromedarius*	Arabian Camel	XP_010992036	205	100
*Vicugna pacos*	Alpaca	XP_006196877.1	205	98.6
*Bos taurus*	Cattle	NP_001179986.1	205	98.1
*Equus przewalskii*	Horse	XP_001504139.1	205	94.6
*Cricetulus griseus*	Chinese hamster	ERE84162.1	209	96.2
*Castor canadensis*	American beaver	XP_020014289.1	209	96.2

**Table 6 ijms-20-02291-t006:** Predicted discontinuous antigenic epitopes of the cLin-28 protein.

Start	End	Peptide	Length	Score	3D Structure
117	134	ERRPKGKNMQKRRSKGDR	18	0.803	A
173	205	KAQQAPSSQGKPAYFREEEEEIHSSAMLPEAQN	33	0.785	B
57	65	RAGVALDPP	9	0.709	C
29	35	EEPQLLH	7	0.702	D
1	6	MGSVSN	6	0.622	E
141	156	LDHHAKECKLPPQPKK	16	0.617	F
163	170	INHMVASC	8	0.574	G
45	48	VRMG	4	0.556	H
94	104	KKSAKGLESIR	11	0.518	I

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
