# Peer review of "Molecular Characterization, Bioinformatic Analysis, and Expression Profile of Lin-28 Gene and Its Protein from Arabian Camel (Camelus dromedarius)"

_ijms, 2019, doi:10.3390/ijms20092291_

Round 1
Reviewer 1 Report
The paper deals with analysis of Lin-28 Gene and its product in camel. It is well made and can be published.
Some proposed changes are as follows.
1. Change the title, you analysed also the gene product, not only the gene.
2. Show data for comparing of nucleotide sequences among species (Fig 3) similarly as for amino acid sequences (Fig 6).
3. The analysis was made only for one animal. You have to emphasize it throughout the text! The analysis must be repeated in more animals.
4. Re-formulate the Conclusions section. Summarize systematically and concisely the results.
5. Formal errors:
- R. 100, what is (Figure ??).
6. R. 104 flexibility; r. 180 The tree...; r. 184 We used....
7. Why have you placed the tables and figures in the text, it is not usual in version for peer-review.
Author Response
1. Molecular Characterization, Bioinformatic Analysis, and Expression Profile of Lin-28 Gene and Its Protein from Arabian Camel (Camelus dromedaries)
2. The figure 4 has been performed.
3. Dear reviewer, many thanks for your point. I have emphasized the study on Camelus Dromedaries in various aspects in this manuscript such as from line 18 to 25 in the introduction section and then compared its nucleotide and protein sequences with nine different animals. Finally, we compared its 3D protein structure with the one in human.
4. The conclusion section has been reformulated and the results have been summarized.
5. has been corrected (Figure 9)
6. Words have been corrected.
Reviewer 2 Report
The paper by Alharbi et al. “Molecular Characterization, Bioinformatic Analysis, and Expression Profile of Lin-28 Gene from Arabian Camel (Camelus dromedarius)” Describes a computational and molecular characterization of the lin-28
First, the authors sequenced and confirmed the identity of the lin-28 protein. Then they show the expression pattern from different tissues. Further, they characterised the lin-28 in terms of its biophysical properties.
Overall, I think the authors have done a good job in confirming the identity of the lin-28 protein from a camel which is what they set out to do in the first place.
Things to improve
1. The authors need to discuss their data more thoroughly as this is not very clear from the present presentation
2. The overall aim and importance of doing the study are not very clear. The authors should try to improve the introduction. In this light, the discussion and conclusion can be clearly made
3. The language has to strongly improve. The authors have to through the manus thoroughly.
4. it is confusing sometimes with the wording used in the text, for example, line 86-87, the authors try to differentiate total charge residues from basic and acidic residues. This is not clear as basic and acidic residues are all charged. if the authors meant the same thing then they have to make it clear. the numbers do not add up.
I will suggest the paper be accepted after a major revision of the language and a better discussion has been added
Author Response
1. The results, discussion, and conclusion sections have been elucidated and reorganized.
2. We believe that this genetic and structural information will be a helpful source for the annotation of the Arabian camel genome. We also assume that the study of biochemical and biophysical properties of cLin-28 gene is likely to provide molecular insights into Arabian camel genome.
4. As the function and structure of any protein depend on its amino acid compositions thus we tried to explore the amino acid properties that make up the clin-28 protein. The amino acids, which are given in table 2, build up the cLin-28 protein (205 amino acids). They can, however, be classified into overlapping groups that share some common physical and chemical properties, such as size and electrical charge. We classified the amino acids as followed: charged(RKHYCDE), acidic(DE), basic(KR), polar(NCQSTY), and Hydrophobic(ALFWV).
Round 2
Reviewer 2 Report
I feel the manuscript has been greatly improved. I will accept the manuscript for publication